# Breast Ultrasound Image Synthesis using Deep Convolutional Generative Adversarial Networks

**DOI:** 10.3390/diagnostics9040176

**Published:** 2019-11-06

**Authors:** Tomoyuki Fujioka, Mio Mori, Kazunori Kubota, Yuka Kikuchi, Leona Katsuta, Mio Adachi, Goshi Oda, Tsuyoshi Nakagawa, Yoshio Kitazume, Ukihide Tateishi

**Affiliations:** 1Department of Diagnostic Radiology, Tokyo Medical and Dental University Hospital 1-5-45, Yushima, Bunkyo-ku, Tokyo 113-8501, Japan; fjokmrad@tmd.ac.jp (T.F.); kbtmrad@tmd.ac.jp (K.K.); 11.ruby.89@gmail.com (Y.K.); leonah@jcom.home.ne.jp (L.K.); ktzmmrad@tmd.ac.jp (Y.K.); ttisdrnm@tmd.ac.jp (U.T.); 2Department of Radiology, Dokkyo Medical University Hospital, 880 Kitakobayashi, Mibu, Shimotsugagun, Tochigi 321-0293, Japan; 3Department of Surgery, Breast Surgery, Tokyo Medical and Dental University Hospital 1-5-45, Yushima, Bunkyo-ku, Tokyo 113-8501, Japan; mioadachi1016@gmail.com (M.A.); oda.srg2@tmd.ac.jp (G.O.); nakagawa.srg2@tmd.ac.jp (T.N.)

**Keywords:** breast imaging, ultrasound, deep learning, convolutional neural network, generative adversarial networks, artificial intelligence

## Abstract

Deep convolutional generative adversarial networks (DCGANs) are newly developed tools for generating synthesized images. To determine the clinical utility of synthesized images, we generated breast ultrasound images and assessed their quality and clinical value. After retrospectively collecting 528 images of 144 benign masses and 529 images of 216 malignant masses in the breasts, synthesized images were generated using a DCGAN with 50, 100, 200, 500, and 1000 epochs. The synthesized (*n* = 20) and original (*n* = 40) images were evaluated by two radiologists, who scored them for overall quality, definition of anatomic structures, and visualization of the masses on a five-point scale. They also scored the possibility of images being original. Although there was no significant difference between the images synthesized with 1000 and 500 epochs, the latter were evaluated as being of higher quality than all other images. Moreover, 2.5%, 0%, 12.5%, 37.5%, and 22.5% of the images synthesized with 50, 100, 200, 500, and 1000 epochs, respectively, and 14% of the original images were indistinguishable from one another. Interobserver agreement was very good (|*r*| = 0.708–0.825, *p* < 0.001). Therefore, DCGAN can generate high-quality and realistic synthesized breast ultrasound images that are indistinguishable from the original images.

## 1. Introduction

With the recent development of deep learning technology, the use of deep learning methods for medical image synthesis has increased dramatically [1,2,3]. One of the most interesting breakthroughs in the field of deep learning is the advent of generative adversarial networks (GANs), which consist of effective machine learning frameworks used to train unsupervised generative models. A GAN is a special type of neural network model in which two networks are trained simultaneously; one focuses on image generation and the other on discrimination [4]. We can apply a GAN to many medical applications when image reconstruction, segmentation, detection, classification, or cross-modality synthesis is necessary [5]. A deep convolutional GAN (DCGAN) is a direct extension of GAN that uses convolutional and transpose–convolutional layers in the discriminator and generator, respectively. DCGANs can reportedly generate high-quality medical images [6]. When using diagnostic images in publications or lectures, patient consent is required and their privacy must be preserved. Moreover, the collection of numerous diagnostic images can be time-consuming [5]. However, these problems may be overcome if synthetic images with the same characteristics as the real image are available using a DCGAN. Unfortunately, the reports on medical imaging which employ a GAN have not demonstrated any clinical evaluations using the images to date, and to the best of our knowledge, there are no reports of breast ultrasound image synthesis using a DCGAN.

To this end, the purpose of this study was to use a DCGAN to generate breast ultrasound images and evaluate their clinical value.

## 2. Materials and Methods

### 2.1. Patients

Our medical ethics committee (Tokyo Medical and Dental University Hospital Ethics Committee) approved the retrospective study in 20 September 2019 and waived the requirement for written informed consent from patients. In the present study, the inclusion criteria were as follows: (a) those who underwent breast ultrasound examination at our hospital between January 2010 and December 2017 and presented with breast masses and (b) those whose masses were diagnosed as being benign or malignant by histopathological examination or >2-year follow-up. The exclusion criteria were as follows: (a) those who received hormone therapy, chemotherapy, or radiation therapy and (b) those who were <20 years of age. A breast radiologist (Tomoyuki Fujioka) and medical student (Mizuki Kimura) randomly selected and extracted the maximum six different cross-sectional images of one mass and the maximum two masses in one patient. Ultrasound images with strong artifacts were excluded.

Five radiologists with 4–20 years of experience performed the ultrasound examinations using an Aplio XG scanner with a PLT-805AT 8.0-MHz linear probe (Toshiba Medical Systems, Tochigi, Japan), an Aplio 500 scanner with a PLT-805AT 8.0-MHz linear probe (Toshiba Medical Systems, Tochigi, Japan), or an EUB-7500 scanner with a EUP-L54MA 9.75-MHz linear probe (Hitachi Medical Systems, Tokyo, Japan). The radiologists acquired static images in the vertical and horizontal planes and measured the maximum diameter of the masses.

### 2.2. Data Set

In this study, all solid and cystic masses, including simple cysts, were evaluated. We used the same set of data that were employed in our previous work [7]. Ultrasound DICOM images were converted to JPEG figures using the viewing software TFS-01 (Toshiba Medical Systems, Tochigi, Japan) and trimmed to include the chest wall with Microsoft Paint (Microsoft, Redmond, WA, USA) for analysis.

Table 1 summarizes the details of patient numbers, masses, images, age, and maximum mass diameter. We extracted a maximum of six cross-sectional images per mass and two masses per patient. For image synthesis, we used a total of 1057 images of 360 masses in 355 patients (528 images of 144 benign masses in 141 patients and 529 images of 216 malignant masses in 214 patients). Table 2 presents the results of histopathological tests of the masses.

### 2.3. Image Synthesis

Image synthesis was performed on DEEPstation DK-1000 (UEI, Tokyo, Japan) containing the graphics processing unit GeForce GTX 1080 (NVIDIA, CA, USA), central processing unit Core i7-8700 (Intel, CA, USA), and graphical user interface-based deep learning tool Deep Analyzer (GHELIA, Tokyo, Japan). Images were constructed using a DCGAN [6]. The DCGAN discriminator was made up of strided convolution layers, batch norm layers, and LeakyReLU activations. The generator comprised of transpose–convolutional layers, batch norm layers, and ReLU activations. The strided transpose–convolutional layers allowed the latent vector to be transformed into a volume with the same shape as an image. The parameters for the generator and discriminator were the same as those reported in the study by Radford et al. [6]: optimizer algorithm = Adam (lr = 0.0002, β1 = 0.5, β2 = 0.999, eps = le^−8^). The image data were set to be input at a pixel size of 256 × 256 and output at a pixel size of 256 × 256. After building the models, we generated 20 images with 50, 100, 200, 500, and 1000 epochs. Moreover, we randomly selected 40 original images. Figure 1 shows five examples of the synthetic and original breast ultrasound images.

### 2.4. Radiologist Readout

In the present study, two breast radiologists [reader 1 (Mio Mori), who had 6 years of experience in breast imaging, and reader 2 (Tomoyuki Fujioka), who had 10 years of experience in breast imaging] assessed the ultrasound images. The radiologists subjectively evaluated the 20 generated images with 50, 100, 200, 500, and 1000 epochs and 40 original images on a scale of 1–5 (1 = excellent, 2 = good, 3 = normal, 4 = poor, and 5 = very poor) for overall image quality, definition of anatomic structures, and visualization of the masses. The quality of the original breast ultrasound image at our facility was used as the standard for score 1. They also score the possibility of the images being original on a scale of 1–5 (1 = 100%, 2 = 75%, 3 = 50%, 4 = 25%, and 5 = 0%).

### 2.5. Statistical Analysis

All statistical analyses were performed using the EZR software package version 1.31 (Saitama Medical Center, Jichi Medical University, Saitama, Japan) [8].

In this work, the data are presented as the mean and standard deviation. Mann–Whitney U-tests were used for the analysis of characteristics, including patient age, maximum mass diameter, and image quality (overall quality of images, definition of anatomic structures, and visualization of the masses) using the mean five-point assessment scores given by readers 1 and 2. The interobserver agreement of scores given by readers 1 and 2 was assessed using Spearman’s coefficient of correlation. A *p* value of <0.05 was considered to be statistically significant.

## 3. Results

Malignant masses were larger than benign masses, and patients with malignant masses were significantly older than those with benign masses (Table 1).

Table 3 and Table 4 summarize the results of the evaluation and comparison of image quality for the synthetic and original images. In the five-point assessment of overall image quality, definition of anatomic structures, and visualization of the masses, the scores for the original images were lower than those for the synthesized images (*p* < 0.001). Although there was no significant difference between images synthesized with 1000 and 500 epochs (*p* = 0.725–1.000), the latter were evaluated as being of better quality than all other images. Images synthesized with 500 epochs were significantly different from those synthesized with 50, 100, or 200 epochs in terms of overall quality (*p* < 0.016) and visualization of the masses (*p* < 0.006). Moreover, images synthesized with 500 epochs were significantly different from those synthesized with 50 or 100 epochs in terms of the definition of anatomic structures (*p* < 0.001). Interobserver agreement was very good for overall quality (|*r*| = 0.825, *p* < 0.001), definition of anatomic structures (|*r*| = 0.784, *p* < 0.001), and visualization of the masses (|*r*| = 0.708, *p* < 0.001). In the evaluation of the possibility of images being originals, 2.5%, 0%, 12.5%, 37.5%, and 22.5% of the images synthesized with 50, 100, 200, 500, and 1000 epochs, respectively, were misinterpreted as being original images (possibility of the synthesized images being original with 100% and 75%) or were indistinguishable from the original images (possibility of the synthesized images being original with 50%). Moreover, 14.0% of the original images were indistinguishable from the synthesized images (possibility of the original images being synthesized with 50%) (Table 5).

## 4. Discussion

In this study, we used a DCGAN to synthesize breast ultrasound images and asked two experienced breast radiologists to evaluate those images from a clinical perspective. Our results indicate that high-quality breast ultrasound images that are indistinguishable from the original images can be generated using the DCGAN.

The concept of adversarial training is relatively new, and great progress has recently been made in this area of research [4]. In our study, the fully connected layer was used as a building block for a GAN, which was later replaced by the complete convolutional downsampling/upsampling layer for the DCGAN [6]. The DCGAN showed better training stability and generated high-quality medical images. GANs have attracted much attention since their development, and the number of reports involving their use has been rising each year [5].

Previously, Beers et al. showed realistic medical images in two different domains, namely retinal fundus photographs showing retinopathy associated with prematurity and two-dimensional magnetic resonance images of gliomas using progressive growing of GANs, which can create photorealistic images at high resolutions [9]. Additionally, Frid-Adar et al. reported that the accuracy of classifying liver lesions was increased using synthetic images generated by a DCGAN from a limited dataset of computed tomography images of 182 liver lesions (including cysts, metastases, and hemangiomas) as a training set for the convolutional neural network [10]. Synthetic images of lung nodules on computed tomography scans, chest infections on X-rays, and breast tissues on mammography using GANs have also been reported [11,12,13]. However, reports of the use of GAN for medical images synthesis have not demonstrated any associated clinical evaluations.

In this study, we generated synthetic images with 50, 100, 200, 500, and 1000 epochs and demonstrated that as the number of learning repetitions increased, the quality of the final image increased. Although there was no significant difference between the images synthesized with 1000 and 500 epochs, the latter showed a better quality than all other images. Therefore, to generate a high-quality synthetic image, multiple epochs are necessary. However, if the number of learning repetitions is very high, overlearning may occur and the accuracy of the generated images may decrease. In this study, we adopted the parameters for the generator and discriminator from our previous work [6] and generated high-quality breast ultrasound synthetic images.

In reference to the synthetic images derived using 500 epochs, 37.5% of the images were misinterpreted as being original or were indistinguishable from the original images and, interestingly, 14.0% of the original images were indistinguishable from the synthesized images. This finding demonstrates that the quality of the realistic and phenotypically diverse synthetic images was such that they perplexed the readers so that they could not confidently assert which images were original.

As research on generated synthetic images using GANs progresses, we may generate realistic data that can be applied to training materials for radiologists and deep learning networks that resolve problems such as the requirement for patient agreement when publishing images and the time and effort required to collect images [5]. The rational that images generated using a DCGAN are acceptable for diagnostic images has not been provided. Synthesized data are still virtual to date, and therefore it is risky to make clinical or scientific conclusions with such data.

This study has several limitations. First, the retrospective study was conducted at a single institution using three ultrasound systems from two companies. Therefore, extensive, multicenter studies are warranted to validate our findings. Second, not all recurrent lesions were diagnosed using cytological or histological diagnoses. Third, we performed this study using images that were converted to 256 × 256 pixel JPEG images. This image processing step might result in loss of information and influence the performance of the generating models. Fourth, the synthetic images were generated using data from a mixture of benign and malignant masses. It may be possible to improve the reliability of the synthetic images by collecting data from more cases, generating classified ultrasound images representing different pathological cases, and comparing original date and patient information. Fifth, the readers subjectively evaluated the generated images and original images; therefore, bias of the evaluation could not be completely removed. Finally, we examined only ultrasound images of breast masses; we have not fully verified whether images of normal breast tissues and non-mass-containing breast lesions can be applied.

## 5. Conclusions

DCGANs can generate high-quality and realistic breast ultrasound images that are indistinguishable from original images.

## Figures and Tables

**Figure 1 diagnostics-09-00176-f001:**
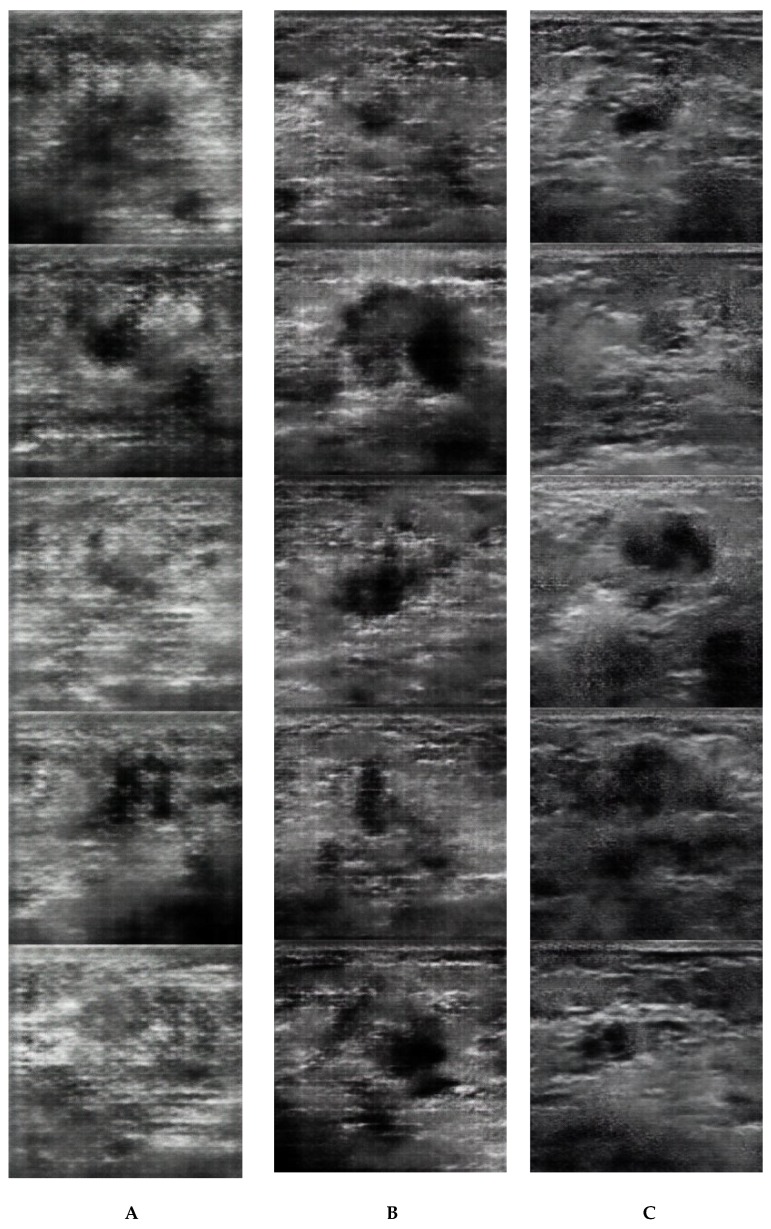
Examples of five synthetic images generated with 50 (**A**), 100 (**B**), 200 (**C**), 500 (**D**), and 1000 (**E**) epochs and the original images (**F**).

**Table 1 diagnostics-09-00176-t001:** Characteristics of patients and their masses.

		Benign	Malignant	*p*
Patients (*n*)		141	214	
Masses (*n*)		144	216	
Images (*n*)		528	529	
Age	Mean (y)	48.8 ± 12.1	60.3 ± 12.6	*p* < 0.001
	Range (y)	21–84	27–84	
Maximum Diameter	Mean (mm)	13.5 ± 8.1	17.1 ± 7.9	*p* < 0.001
	Range (mm)	4–50	5–41	

Comparison was performed using the Mann–Whitney U-test.

**Table 2 diagnostics-09-00176-t002:** Histopathology of masses.

Benign (*n* = 144)	Malignant (*n* = 216)
Fibroadenoma 47	Ductal Carcinoma in Situ 17
Mastopathy 19	Invasive Ductal Cancer 168
Intraductal Papilloma 17	Invasive Lobular Carcinoma 9
Phyllodes Tumor (Benign) 2	Mucinous Carcinoma 8
Fibrous Disease 1	Apocrine Carcinoma 7
Lactating Adenoma 1	Invasive Micropapillary Carcinoma 2
Abscess 1	Malignant Lymphoma 1
Adenosis 1	Medullary Carcinoma 1
Pseudoangiomatous Stromal Hyperplasia 1	Adenoid Cystic Carcinoma 1
Radial scar/Complex Sclerosing Lesion 1	Phyllodes Tumor (Malignant) 1
No Malignancy 5	Adenomyoepithelioma With Carcinoma 1
Not Known 48 (Diagnosed by Follow-up)	

**Table 3 diagnostics-09-00176-t003:** Five-point assessment score of synthetic and original images.

	Overall Quality of Images			Definition of Anatomic Structures			Visualization of the Masses		
Reader	R1	R2	R1&2	R1	R2	R1&2	R1	R2	R1&2
50 epochs	4.85 ± 0.37	4.25 ± 0.72	4.55 ± 0.48	4.85 ± 0.37	4.20 ± 0.89	4.53 ± 0.53	4.30 ± 0.57	4.05 ± 0.69	4.18 ± 0.54
100 epochs	4.55 ± 0.51	4.05 ± 0.51	4.30 ± 0.41	4.80 ± 0.41	4.00 ± 0.65	4.40 ± 0.38	4.15 ± 0.49	3.50 ± 0.69	3.83 ± 0.49
200 epochs	4.10 ± 0.64	3.00 ± 0.92	3.55 ± 0.63	4.05 ± 0.60	2.95 ± 0.83	3.50 ± 0.54	4.25 ± 0.85	2.85 ± 0.88	3.55 ± 0.71
500 epochs	3.45 ± 0.83	2.35 ± 0.59	2.90 ± 0.55	3.50 ± 0.83	2.55 ± 0.60	3.03 ± 0.60	3.45 ± 0.83	2.10 ± 0.45	2.78 ± 0.41
1000 epochs	3.70 ± 0.92	2.65 ± 0.93	3.18 ± 0.73	4.00 ± 0.86	2.80 ± 0.70	3.40 ± 0.64	3.40 ± 0.94	2.15 ± 0.99	2.78 ± 0.85
Real	1.48 ± 0.60	1.20 ± 0.41	1.34 ± 0.40	1.45 ± 0.64	1.38 ± 0.59	1.41 ± 0.44	1.55 ± 0.75	1.35 ± 0.62	1.45 ± 0.53

R1:Reader1, R2:Reader2, R1&2: Average of Reader1 and 2.

**Table 4 diagnostics-09-00176-t004:** Comparison of quality of synthetic and original images.

	Overall Quality of Images (*p*)	Definition of Anatomic Structures (*p*)	Visualization of the Masses (*p*)
50 vs. 100	0.470	1.000	0.426
50 vs. 200	<0.001	<0.001	0.100
50 vs. 500	<0.001	<0.001	<0.001
50 vs. 1000	<0.001	<0.001	<0.001
50 vs. Real	<0.001	<0.001	<0.001
100 vs. 200	<0.001	<0.001	1.000
100 vs. 500	<0.001	<0.001	<0.001
100 vs. 1000	<0.001	<0.001	<0.001
100 vs. Real	<0.001	<0.001	<0.001
200 vs. 500	0.016	0.095	0.016
200 vs. 1000	0.897	1.000	0.071
200 vs. Real	<0.001	<0.001	<0.001
500 vs. 1000	1.000	0.725	1.000
500 vs. Real	<0.001	<0.001	<0.001
1000 vs. Real	<0.001	<0.001	<0.001

Average score of Reader1 and 2; Mann–Whitney U-test was performed.

**Table 5 diagnostics-09-00176-t005:** Evaluation for possibility of original images.

Possibility of Original IMAGES	50 epochs		100 epochs		200 epochs		500 epochs		1000 epochs		Real	
Score	R1	R2	R1	R2	R1	R2	R1	R2	R1	R2	R1	R2
1 (100%)	0	0	0	0	0	0	0	0	0	0	18	25
2 (75%)	0	0	0	0	0	2	1	2	1	1	13	13
3 (50%)	0	1	0	0	1	2	2	10	2	5	9	2
4 (25%)	0	4	0	8	3	15	10	8	5	11	0	0
5 (0%)	20	15	20	12	16	1	7	0	12	3	0	0
Indistinguishable Images (%) (Average of R1&R2)	0%	5%	0%	0%	5%	20%	15%	60%	15%	30%	23%	5%
	(2.5%)		(0%)		(12.5%)		(37.5%)		(22.5%)		(14%)	

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
