# Peer review of "Breast Ultrasound Image Synthesis using Deep Convolutional Generative Adversarial Networks"

_diagnostics, 2019, doi:10.3390/diagnostics9040176_

Round 1

Reviewer 1 Report

The purpose of the study reported in this paper is the evaluation by 2 radiologists of the quality and clinical value of simulated ultrasound (US) images of the breast with benign and malignant lesions. The simulated images are generated using deep convolutional generative adversarial networks (DCGANs). The evaluation of the image quality is performed subjectively using scores. The authors conclude that DCGAN can generate high-quality and realistic synthesized breast US images. Applications could be the use of the synthesized images in publications (patient consent is required) and the collection of large amount of images (for ex. for the training of machine learning algorithms).

The paper is short and well written. The methods, results and conclusions are clearly described. The topic of this study is relevant and the conclusions interesting. Only few points remain unclear to me. They are described in the following. Moreover, one limitation of this study is the subjective evaluation of the images by the radiologists. Could the authors suggest ideas to perform a more objective evaluation? Otherwise, I recommend this paper for publication after minor revision.

Section 2.1 l. 58-60: What do you mean with “A breast radiologist (T.F.) and medical student (M.K.) randomly selected one patient’s breast mass for this study from a database of radiological reports and clinical records at our facility.”? What is the purpose? Why only “one patient’s breast mass”?

Section 2.2 table 1: Are the P-values computed between the ages and mass volumes of patients with benign and malignant masses relevant for the study? If this is the case, please explain why. If not, I suggest to remove the values from table.

Section 3 l. 116-117: Again, in which scope the information “Malignant masses were larger than benign masses, and patients with malignant masses were significantly older than those with benign masses (Table 1).” is relevant for this study?

Section 3 table 4: Which values are reported in Table 4? The P-values? If yes, could you please provide 3 digits after the . (point) as it is done in the text? This is otherwise confusing.

Section 3 l. 129 to 133: For better clarity I would add: “In the evaluation of the possibility of images being originals, 2.5%, 0%, 12.5%, 37.5%, and 22.5% of the images synthesized with 50, 100, 200, 500, and 1000 epochs, respectively, were misinterpreted as being original images (possibility of the synthesized images being original with 100% and 75%) or were indistinguishable from the original images (possibility of the synthesized images being original with 50%). Moreover, 14.0% of the original images were indistinguishable from the synthesized images (possibility of the original images being synthesized with 50%) (Table 5).”, or something like that.

Section 3 table 5: Please check the “Indistinguishable Masses (%)” of R1 and R2 for 500 epochs. It seems that the 60% is for R2 and the 15% for R1. Moreover I would prefer the term of “Indistinguishable Images (%)” rather than “Indistinguishable Masses (%)” which is too extrapolated.

Section 4 l. 170: “interestingly, 22.5% of the original images were indistinguishable from the synthesized images.” Do you mean “14% of the original images”?

Section 4: Could the authors suggest ideas to perform a more objective evaluation of the synthesized images? For ex. synthesized images of benign and malignant masses could be separately generated. The mass volumes could be measured and compared to the values measured in the original images.

Author Response

We wish to express our appreciation to the Reviewer for his or her insightful comments, which have significantly helped us to improve the paper.

Reviewer #1: The purpose of the study reported in this paper is the evaluation by 2 radiologists of the quality and clinical value of simulated ultrasound (US) images of the breast with benign and malignant lesions. The simulated images are generated using deep convolutional generative adversarial networks (DCGANs). The evaluation of the image quality is performed subjectively using scores. The authors conclude that DCGAN can generate high-quality and realistic synthesized breast US images. Applications could be the use of the synthesized images in publications (patient consent is required) and the collection of large amount of images (for ex. for the training of machine learning algorithms).

The paper is short and well written. The methods, results and conclusions are clearly described. The topic of this study is relevant and the conclusions interesting. Only few points remain unclear to me. They are described in the following. Moreover, one limitation of this study is the subjective evaluation of the images by the radiologists. Could the authors suggest ideas to perform a more objective evaluation? Otherwise, I recommend this paper for publication after minor revision.

Response: We thank the Reviewer for this pertinent comment.

Section 2.1 l. 58-60: What do you mean with “A breast radiologist (T.F.) and medical student (M.K.) randomly selected one patient’s breast mass for this study from a database of radiological reports and clinical records at our facility.”? What is the purpose? Why only “one patient’s breast mass”?

Response:

We changed as "A breast radiologist (T.F.) and medical student (M.K.) randomly selected and extracted the maximum 6 different cross-sectional images of one mass and the maximum 2 masses in one patient."

Section 2.2 table 1: Are the P-values computed between the ages and mass volumes of patients with benign and malignant masses relevant for the study? If this is the case, please explain why. If not, I suggest to remove the values from table.

Section 3 l. 116-117: Again, in which scope the information “Malignant masses were larger than benign masses, and patients with malignant masses were significantly older than those with benign masses (Table 1).” is relevant for this study?

Response:

This is because we wanted to show age and size distribution and differences in malignant and benign tumors as a reference for generating synthetic images.

Section 3 table 4: Which values are reported in Table 4? The P-values? If yes, could you please provide 3 digits after the. (point) as it is done in the text? This is otherwise confusing.

Response:

We corrected Table 4 according to the reviewer1 instructions.

Section 3 l. 129 to 133: For better clarity I would add: “In the evaluation of the possibility of images being originals, 2.5%, 0%, 12.5%, 37.5%, and 22.5% of the images synthesized with 50, 100, 200, 500, and 1000 epochs, respectively, were misinterpreted as being original images (possibility of the synthesized images being original with 100% and 75%) or were indistinguishable from the original images (possibility of the synthesized images being original with 50%). Moreover, 14.0% of the original images were indistinguishable from the synthesized images (possibility of the original images being synthesized with 50%) (Table 5).”, or something like that.

Response:

 I am grateful for the appropriate advice. I corrected it as suggested by reveiewer1.

Section 3 table 5: Please check the “Indistinguishable Masses (%)” of R1 and R2 for 500 epochs. It seems that the 60% is for R2 and the 15% for R1. Moreover I would prefer the term of “Indistinguishable Images (%)” rather than “Indistinguishable Masses (%)” which is too extrapolated.

Response:

We checked and corrected the numbers.

We changed “Indistinguishable Masses (%)”to “Indistinguishable Images (%)”.

Section 4 l. 170: “interestingly, 22.5% of the original images were indistinguishable from the synthesized images.” Do you mean “14% of the original images”?

Response:

Yes, I do. I changed “interestingly, 14.0% of the original images were indistinguishable from the synthesized images.”

Section 4: Could the authors suggest ideas to perform a more objective evaluation of the synthesized images? For ex. synthesized images of benign and malignant masses could be separately generated. The mass volumes could be measured and compared to the values measured in the original images.

Response:

In this study, “The quality of the original breast ultrasound image at our facility were used as the standard for score 1 of overall image quality, definition of anatomic structures, and visualization of the masses.” We add this sentence in 2. Materials and Methods - 2.4. Radiologist readout. However, the bias of the evaluation could not be completely removed, so we add that to the limitation as “Fifth, the readers subjectively evaluated the generated images and original images; therefore, the bias of the evaluation could not be completely removed.”

In the next study, I would like to do to generate classified ultrasound images representing different pathological cases using more training data.

We add“In this study synthetic images were generated with data with a mixture of benign and malignant masses. It may be possible to improve the reliability of the synthetic images by collecting more cases, generating classified ultrasound images representing different pathological cases and comparing original date and patient information. ” in 4. Discussion.

Reviewer 2 Report

The authors introduce a deep learning-based approach to generate synthesized ultrasound images. The generated images were evaluated by two radiologists and the results suggested that the images generated from DCGEN were comparable and some were indistinguishable from the original training ultrasound images. While the approach is interesting, I have following concerns before publishing this journal article.

The goal of medical image synthesis is not well defined.

The citations 1-3 present the review of current usage of deep learning for medical imaging. However, the motivation for synthesizing the virtual ultrasound images is not explained, which significantly reduce the impact of the paper. The limited description is as follow:

“When using diagnostic images in publications or lectures, patient consent is required and their privacy must be preserved. Moreover, the collection of numerous diagnostic images can be time-consuming. These problems can, however, be overcome if images generated using DCGAN are available.”

The problems of this statement are 1) the rational that the images generated using DCGAN are acceptable over diagnostic images was not provided. Synthesized data are still virtual date, and it is riskly to make clinical or scientific conclusion with those data. 2) The data synthesis is not possible with any universal patient data. How can the author ensure that the specific pathological case can be simulated through deep learning? 3) Ultrasound images are prone to orientation and contact pressure. Training the data without these information could introduce artifacts. 4) How significant the lack of patient data is?  

How the readers were able to provide the score?

For “quality of images”, it is not clearly defined what the readers were focusing on when providing the score. How to eliminate the bias of the evaluation? The reader should not be used to read the artificial ultrasound images, and it is hard to find what would be the reasonable conclusion.

Results do not correctly support the conclusion because the goal is not correctly defined.

While the authors concluded that some artificial images were indistinguishable, from Table 3, it is obvious that the image appearance of DCGAN images were different from real ones. The reviewer suspects that having partial successful images do not support the hypothesis that the synthesized images can replace the real clinical images. It is hard to justify the innovation of the paper if the proof of this hypothesis is failed.

The images appeared in Figure 1 were very noisy and the resolution seems degraded. It may be useful to use a research platform that provides raw ultrasound data (RF data or channel data). The reviewer wonders if the DCGAN engine understands what type of pathological data will be generated in each case. With more data and well-defined training data sets, it may be possible to generate classified ultrasound images representing different pathological cases.

Author Response

We wish to express our appreciation to the Reviewer for his or her insightful comments, which have significantly helped us to improve the paper.

The authors introduce a deep learning-based approach to generate synthesized ultrasound images. The generated images were evaluated by two radiologists and the results suggested that the images generated from DCGEN were comparable and some were indistinguishable from the original training ultrasound images. While the approach is interesting, I have following concerns before publishing this journal article.

Response: We thank the Reviewer for this pertinent comment.

The goal of medical image synthesis is not well defined.

The citations 1-3 present the review of current usage of deep learning for medical imaging. However, the motivation for synthesizing the virtual ultrasound images is not explained, which significantly reduce the impact of the paper. The limited description is as follow:

“When using diagnostic images in publications or lectures, patient consent is required and their privacy must be preserved. Moreover, the collection of numerous diagnostic images can be time-consuming. These problems can, however, be overcome if images generated using DCGAN are available.”

The problems of this statement are 1) the rational that the images generated using DCGAN are acceptable over diagnostic images was not provided. Synthesized data are still virtual date, and it is riskly to make clinical or scientific conclusion with those data. 2) The data synthesis is not possible with any universal patient data. How can the author ensure that the specific pathological case can be simulated through deep learning? 3) Ultrasound images are prone to orientation and contact pressure. Training the data without these information could introduce artifacts. 4) How significant the lack of patient data is?

Response 1):

“These problems can, however, be overcome if images generated using DCGAN are available.” is too strong expression and needs further consideration.

We changed as“However, these problems may be overcome if synthetic images with the same characteristics as the real image are available using DCGAN.” and add“The rational that the images generated using DCGAN are acceptable over diagnostic images was not provided. Synthesized data are still virtual date, and it is risky to make clinical or scientific conclusions with those data.” in 4. Discussion.

Response 2.4):

“The synthetic images were generated using data of a mixture of benign and malignant masses. It may be possible to improve the reliability of the synthetic images by collecting data of more cases, generating classified ultrasound images representing different pathological cases, and comparing original date and patient information.” We add this sentence in limitation - 4. Discussion.

Response 3):

 Since images with strong artifacts are excluded when selecting images, we think the effect may be small. We add “Ultrasound images with strong artifacts are excluded.” in 2. Materials and Methods - 2.1. Patients 

How the readers were able to provide the score?

For “quality of images”, it is not clearly defined what the readers were focusing on when providing the score. How to eliminate the bias of the evaluation? The reader should not be used to read the artificial ultrasound images, and it is hard to find what would be the reasonable conclusion.

 Response:

In this study, “The quality of the original breast ultrasound image at our facility was used as the standard for score 1 of overall image quality, definition of anatomic structures, and visualization of the masses.” We add this sentence in 2. Materials and Methods - 2.4. Radiologist readout.

However, the bias of the evaluation could not be completely removed, so we add that to the limitation as “Fifth, the readers subjectively evaluated the generated images and original images; therefore, the bias of the evaluation could not be completely removed.”

Results do not correctly support the conclusion because the goal is not correctly defined.

While the authors concluded that some artificial images were indistinguishable, from Table 3, it is obvious that the image appearance of DCGAN images were different from real ones. The reviewer suspects that having partial successful images do not support the hypothesis that the synthesized images can replace the real clinical images. It is hard to justify the innovation of the paper if the proof of this hypothesis is failed.

 Response:

As shown in Figure 1, you can see the difference in image quality by arranging each epochs and real image, but if you arrange them all at random, you cannot distinguish some high quality synthetic images and original images.

I would like you to understand that this is an interesting and important point of this study.

The images appeared in Figure 1 were very noisy and the resolution seems degraded. It may be useful to use a research platform that provides raw ultrasound data (RF data or channel data). The reviewer wonders if the DCGAN engine understands what type of pathological data will be generated in each case. With more data and well-defined training data sets, it may be possible to generate classified ultrasound images representing different pathological cases.

 Response:

The synthetic images generated using DCGAN with the data of our facility is as shown in "Figure 1". Although the quality of images with low epochs is not good, it has been found that the image quality improves with increasing epochs.

In the next study, I would like to do to generate classified ultrasound images representing different pathological cases using the database the reviewer 2 introduced.

“Fourth, the synthetic images were generated using data of a mixture of benign and malignant masses. It may be possible to improve the reliability of the synthetic images by collecting data of more cases, generating classified ultrasound images representing different pathological cases, and comparing original date and patient information.” We add this sentence in limitation - 4. Discussion.

Round 2

Reviewer 2 Report

The reviewer thinks that the updated manuscript is appropriate to be published in the Diagnostics Journal.